# Understanding and managing new risks on the Nile with the Grand Ethiopian Renaissance Dam

Kevin G. Wheeler [1], Marc Jeuland [2], Jim W. Hall [1], Edith Zagona [3] & Dale Whittington [4,5]✉

When construction of the Grand Ethiopian Renaissance Dam (GERD) is completed, the Nile will have two of the world's largest dams—the High Aswan Dam (HAD) and the GERD—in two different countries (Egypt and Ethiopia). There is not yet agreement on how these dams will operate to manage scarce water resources. We elucidate the potential risks and opportunities to Egypt, Sudan and Ethiopia by simulating the filling period of the reservoir; a *new normal* period after the reservoir fills; and a severe multi-year drought after the filling. Our analysis illustrates how during filling the HAD reservoir could fall to levels not seen in recent decades, although the risk of water shortage in Egypt is relatively low. The *new normal* will benefit Ethiopia and Sudan without significantly affecting water users in Egypt. Management of multi-year droughts will require careful coordination if risks of harmful impacts are to be minimized.

[1] Environmental Change Institute & Oxford Martin School, University of Oxford, Oxford, UK. [2] Sanford School of Public Policy & Duke Global Health Institute, Duke University, Durham, NC, USA. [3] CADSWES, University of Colorado, Boulder, CO, USA. [4] University of North Carolina, Chapel Hill, NC, USA. [5] University of Manchester, Manchester, UK. ✉email: Dale_Whittington@unc.edu

With the construction of the Grand Ethiopian Renaissance Dam (GERD) underway near the Ethiopia-Sudan border, a complex transboundary water situation is at hand: two large dams—the GERD and the High Aswan Dam (HAD)—in two countries, Ethiopia and Egypt, will coexist on a single river—the Nile—with no specific agreement yet on water sharing or reservoir operations. Both reservoirs can capture more than the average annual flow at their respective locations and can thus dramatically change the river's flow. Although 85% of Nile waters originate in Ethiopia, nearly all consumptive use occurs downstream in Egypt and Sudan. Ethiopia's main purpose for building the GERD is power generation rather than consumptive use, but the GERD operations will change downstream flow patterns significantly. This alteration is raising concern about risks of water shortage, which underlines the importance of reaching an agreement on the river's management.

The hydrology of the Nile has been studied for decades[1]; it is characterized by high inter-annual variability, stark differences in geography and climate, and flows modified by natural features and water infrastructure[2] (Fig. 1). Heavy rainfall over Ethiopia from June through September creates highly seasonal flows in the Blue Nile and Atbara tributaries. Rainfall over the equatorial lakes is greatest from March until May, and September until December. Combined with the buffering effect of the Sudd wetlands in South Sudan, this bimodal rainfall pattern results in relatively steady year-round flow in the White Nile. A naturalized hydrologic reconstruction at Aswan from 1900 to 2018[3,4] indicates a range of annual flows between 45.6 billion cubic meters (bcm) and 120 bcm with an average annual flow of 86.5 bcm (Fig. 2). The Blue Nile contributes ~55% of this flow, with the remaining 32% and 13% from the White Nile and Atbara, respectively.

The Nile has been used since antiquity for domestic use, irrigation, and navigation, but large-scale planning and development only began in the late 19th century as British colonial officials proposed a vision of basin-wide management[5]. Ethiopia, however, remained independent and was not party to these aspirations[6]. As countries gained independence, Egypt and Sudan asserted claims to the Nile in their 1959 "Agreement for the Full Utilization of Nile Waters". Not being signatories, Ethiopia and other upstream states do not recognize this agreement[7].

The 1959 Agreement initiated the construction of the HAD with a total storage volume of 162 bcm. Since its construction, sediment accumulation has reduced the capacity by ~7 bcm, with 46% deposited in the live storage. The primary purpose of the HAD has been to meet Egypt's agricultural, municipal, and industrial water requirements through regular annual releases of 55.5 bcm. In practice, releases have often exceeded this amount. Based on operational policies, releases may also be reduced below 55.5 bcm under an existing Drought Management Policy (DMP) if storage falls below 60 bcm (159.4 m above sea level or masl)[8]. According to the 1959 Agreement, 18.5 bcm would be accessible each year to meet Sudanese demands, and 10 bcm was allotted for evaporation loss from the HAD Reservoir.

Ethiopia has long had development ambitions of its own. In 1964, a study of development opportunities on the Nile within Ethiopia highlighted the potential for hydropower generation and irrigation[9]. Following the completion of several comparatively smaller hydropower structures in the Nile tributaries[10], Ethiopia embarked upon the construction of the GERD on the Blue Nile in 2011. With an installed capacity of 5150 MW, the dam is expected to annually generate around 16 TWh of energy[11]. When completed and operational, the GERD will be the largest hydroelectric power generation facility in Africa and the fifth largest in the world. The GERD Reservoir will have a total storage of 74 bcm, 59 bcm of it active, or nearly 1.2 times the average annual flow of the Blue Nile at the dam site. Additional descriptions of the

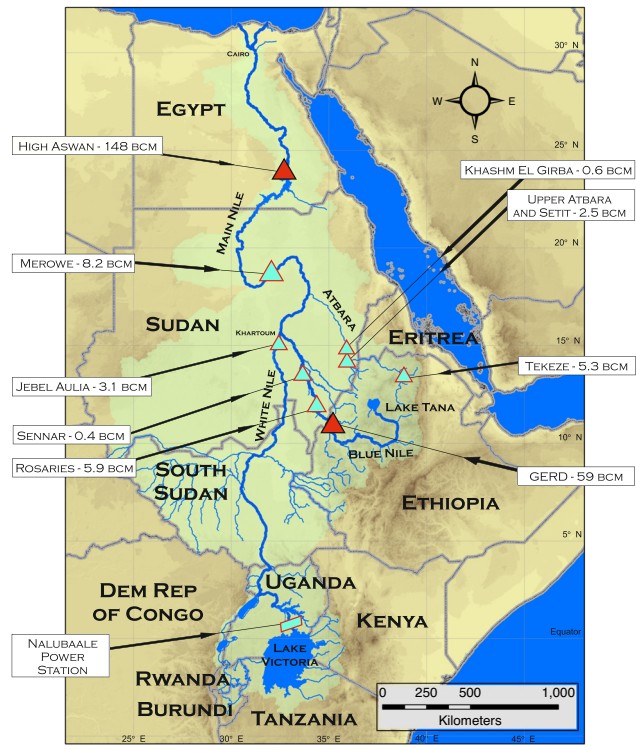

**Fig. 1 Map of the Nile Basin with major infrastructure.** Active reservoir storage volumes are shown for each location.

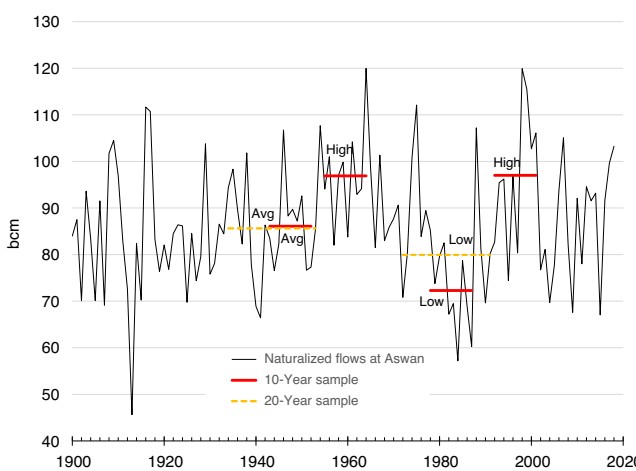

**Fig. 2 Naturalized historical total annual flows at Aswan.** Horizontal bars show 10- and 20-year sequences used in simulations.

GERD and the HAD—as well as other important Eastern Nile infrastructures—are included in the Supplementary Information.

The GERD project has strained relations between the Nile countries. Downstream countries worry about management of the initial period of reservoir filling and have also expressed concern about long-term operations. Both issues have been the subject of difficult, ongoing negotiations between Egypt, Ethiopia, and Sudan. Previous studies of the GERD and HAD operations have sought to identify tradeoffs between different riparian objectives, and to understand the risks and rewards of basin-wide cooperation[12–23]. These efforts provide important insights that will inform multilateral agreements and create a structure for addressing the significantly different interests in the basin. The aim of this paper is to draw attention to critical situations in

which water risks may become severe and potentially socially disruptive.

As future flows in the Nile cannot be predicted with certainty, it has become common practice in reservoir simulation models to stress test systems using synthetic flows, sometimes incorporating future changes that are projected from climate models[24]. However, results from these simulation studies are often not considered sufficiently credible by decision makers[25,26]. Fortunately, the Nile has a long record of flows, and thus we are able to select historical flow sequences that are representative of low, average and high flows, in order to illustrate how the system would perform in these hydrological conditions (see Methods).

We organize our discussion into three stylized eras: (1) the period of filling of the GERD Reservoir; (2) a *new normal* that may begin after filling the reservoir is complete, but during which no severe multi-year drought occurs; and (3) a post-filling period that includes a severe multi-year drought. The periods are stylized because we cannot know with any certainty when they will begin or end, nor the specific hydrology that will accompany them. For each of the three eras, we consider reservoir operations from the perspectives of Egypt, Sudan, and Ethiopia, quantify the risks of water deficits, and discuss potential government and civil society perceptions of the risks that might arise.

## Results

**Filling the GERD reservoir**. The first 'era' is the period in which the reservoir will be filled. As of August 2020, construction of the GERD was over 70% percent complete, and Ethiopia has completed the first year filling of the reservoir by impounding 4.9 bcm. Throughout the filling process, a portion of the inflows into the GERD Reservoir will be retained, but the remainder will be released through turbines to generate hydropower or through spillway and bypass structures[27]. During this period, levels in the HAD Reservoir will decline relative to where they would have been without the GERD, and HAD hydropower production will correspondingly decrease due to reduced hydraulic head on the HAD turbines. Once the GERD reservoir is filled, the average release from the GERD will be equal to its average annual inflow (49 bcm), less annual evaporation losses of ~1.7 bcm, but the seasonal pattern of those releases will have changed.

The downstream consequences of filling the GERD are difficult to assess because they will depend on six key factors: (1) rainfall and flow in the basin during filling; (2) how quickly Ethiopia fills the GERD Reservoir; (3) how hydropower is generated at the GERD, which, in turn, depends on developments in the regional power grid; (4) how the GERD Reservoir filling influences Sudan's withdrawals; (5) the initial storage level in the HAD Reservoir (as of 24 August 2020, it was nearly full at 178.4 masl); and (6) how the HAD is operated during the filling period. There is complexity in each of these factors, which is discussed further in the Methods and Supplementary Information. Here we present results for one filling proposal suggested in 2019 by the National Independent Scientific Research Group[28] with a set of assumptions about each of these elements, as detailed in the Methods section. Although more robust, dynamic filling policies, such as back-up releases from reservoirs during drought[20] or dynamic allocations based on economic benefits[29], could further reduce risks and increase benefits, our aim is to elucidate the benefits and risks associated with filling, and then surmise the societal responses in Era 1 under a variety of conditions. We believe this filling proposal identified by the National Independent Scientific Research Group is sufficient for these purposes.

The results in Figs. 3–7 show: (1) what the storage of the HAD Reservoir would have been if the GERD had not been built; (2) the storage of the HAD Reservoir with the GERD in place; (3) the

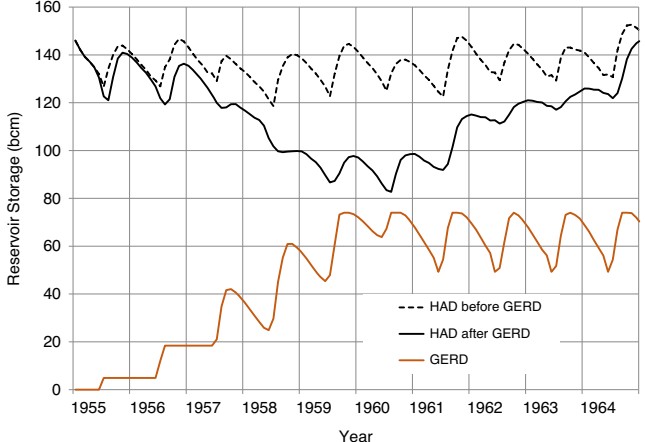

**Fig. 3 GERD filling effects beginning with a historically wet 10-year sequence.** Storage volumes for the GERD and HAD are shown.

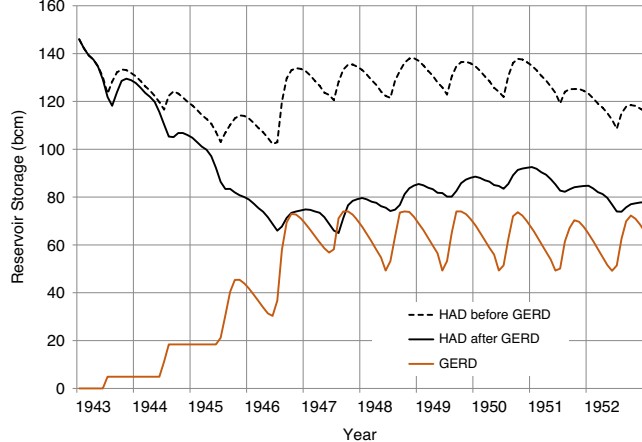

**Fig. 4 GERD filling effects under a historically near-average 10-year sequence.** Storage volumes for the GERD and HAD are shown.

storage of the GERD Reservoir; and (4) when present, the magnitude of annual deficits in water availability in Egypt before and after the GERD, measured as shortfalls relative to a 55.5 bcm/year planned release.

If the reservoir system is subject to the flows experienced in the 1955–1964 flow sequence of wet years, filling the GERD poses no problems for Egypt (Fig. 3 and Supplementary Table 4). The GERD Reservoir transitions from empty to full in five years and subsequently maintains a high, stable release pattern over the remainder of the simulation period. The pool elevation of the HAD never falls below 60 bcm and the storage of the HAD Reservoir recovers quickly. Egypt would not need to invoke the HAD Drought Management Policy (DMP) which would impose restrictions on releases. During these wet conditions, Egypt can release 55.5 bcm in every year of the simulation.

During the 1943–1952 sequence of average years, Egypt is also predicted to be able to deliver 55.5 bcm each year and would not need to invoke the HAD Drought Management Policy (DMP) (Fig. 4 and Supplementary Table 4). The GERD is filled in the 5th year and the HAD Reservoir thereafter experiences a gradual recovery, reaching 78 bcm in storage at the end of 10 years, substantially less than its initial storage of 146 bcm. The results of other similar near-average sequences are somewhat sensitive to the particular pattern in their flows, but the critically low HAD

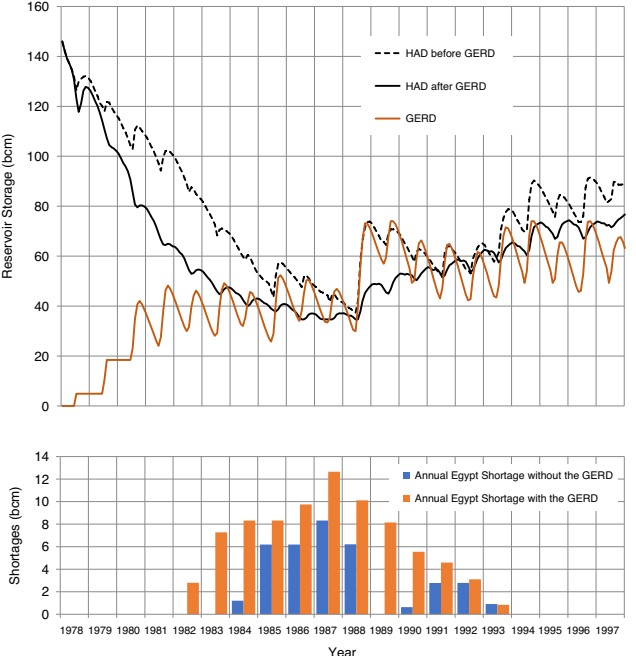

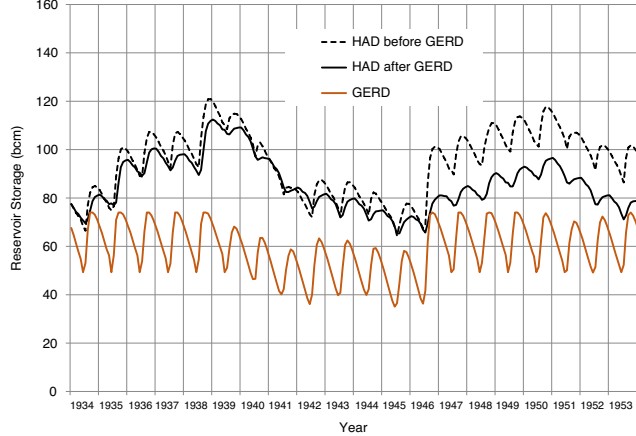

**Fig. 5 GERD filling effects during a historically dry 10-year sequence followed by a recovery period.** Storage volumes for the GERD and HAD are shown above and shortages to Egyptian planned releases are shown below.

**Fig. 6 *New normal* conditions with the GERD under a historically average 20-year sequence.** Storage volumes for the GERD and HAD are shown.

Reservoir level of 147 masl (34.7 bcm of storage) is never reached. An important factor in avoiding such critical levels is the high current storage in the HAD Reservoir. This makes it much less likely that storage behind the HAD will fall to levels that lead to significant restrictions in releases under the current DMP in Egypt. However, the HAD Reservoir is highly likely to fall to levels not seen in recent decades.

The effect of extremely dry conditions such as the 1978–1987 drought during Era 1 is shown in Fig. 5 and Supplementary Table 4, for which the analysis is extended to show how this effect persists over a period of 20 years. Even without the GERD and with the HAD Reservoir starting full, HAD Reservoir storage would fall near to the minimum operation level of 147 masl. All three tiers of Egypt's DMP would be invoked, resulting in a cumulative shortage of 35 bcm over the drought period. Figure 5 also demonstrates how such a long-term drought would have adverse consequences for both Egypt and Ethiopia while the GERD Reservoir is filling. In the first 6 years of the simulation, storage in the HAD Reservoir is predicted to fall to 47 bcm, whilst the GERD has only stored 46 bcm. Deficits in Egypt are predicted in year 5 (2 years earlier than without the GERD). The cumulative additional deficit to Egypt as a result of the GERD over the entire 20-year period is 46 bcm, so the 20-year cumulative deficit due to the combined drought and the GERD filling reaches 81 bcm.

The probability of a re-occurrence of a multi-year drought as severe as 1980s sequence during the GERD filling period is low, but such an event would have a greater impact today than it did then because withdrawals upstream of the HAD are higher today. Despite the low likelihood of such a sequence, the fact that multi-year droughts have occurred from time to time in the past means that careful and cooperative advanced planning is needed to prepare for the possibility that large and persistent deficits could occur.

**The *new normal* after the filling the GERD reservoir.** The second 'era' begins once the GERD Reservoir reaches its full supply level (FSL) and normal operations begin. At that point, the

average annual volume entering the reservoir, less evaporation losses, will flow downstream through the GERD's turbines whenever possible—i.e. during normal, wet conditions, and minor droughts. The variability of flows downstream of the GERD will decrease significantly, as the intra-annual timing of releases is modified to achieve power generation goals at the GERD (whether for baseload or peak power production) and capturing peak flows in the Blue Nile. Net evaporation losses from the GERD Reservoir will average ~1.7 bcm per year and will not fluctuate much from year to year. These losses from the GERD Reservoir will be partially offset by 1.1 bcm of reduced evaporation in the HAD Reservoir when the system reaches a new equilibrium (see Supplementary Note 4 for further discussion of evaporation losses).

To illustrate conditions in the *new normal* era, we start our simulation with both reservoirs at normal operating conditions (total storage of 79.6 bcm including 7 bcm of sediment in the HAD Reservoir [165.5 masl]), and total storage in the GERD of 70.4 bcm [637.9 masl]). As discussed in the Methods, we assume that the GERD is operated to produce a baseload of 1600 MW whenever possible[21], and that target diversions remain similar to those of the filling era, except for modest increases in Ethiopia[10]. We select the 20-year historical sequence from 1934 to 1953 to simulate outcomes that we consider to be broadly representative of a typical sequence of low and high years, but without extreme multi-year drought conditions.

As shown (Fig. 6 and Supplementary Table 5), for an average 20-year sequence that is broadly representative of typical low and high years (1934–1953), the GERD is able to maintain steady releases over time, and storage never falls to the minimum operating level of 18.4 bcm (595 masl). Moreover, the GERD buffers the variability in HAD Reservoir storage relative to the case without the GERD, resulting in lower peaks and periods of higher minimum storage. The HAD Reservoir remains above 60 bcm of storage, and Egypt's DMP is never deployed.

The simulation also demonstrates that storage in the HAD Reservoir is typically lower with the GERD than without it. This should be expected due to the evaporative losses from the GERD and additional evaporation from Sudanese reservoirs, which will likely be operated at higher levels because the Nile flows will be less variable. Both factors will reduce inflows to the HAD Reservoir. While lower levels in the HAD Reservoir may cause people in Egypt to feel that their water supply is less secure, under normal or wet flow conditions the increased predictability of inflows and the continuation of current management approaches at the HAD should enable Egypt to release 55.5 bcm annually

almost all the time even without coordinated operation with the GERD.

Moreover, Sudan will clearly be better off in Era 2 because GERD operations will smooth Blue Nile flows, eliminating flood losses, increasing hydropower generation, decreasing sediment load to the reservoirs and canals, and, most importantly, increasing water for summer irrigation in the Gezira Scheme and other irrigated areas along the Blue Nile[30],[31].

**The consequences of a future severe multi-year drought**. The third 'era' will begin when a sequence of very low flows occurs in the Nile Basin[32]. The probability, severity, and timing of specific sequences of low flows are unknowable, especially as climate change unfolds. For example, a severe multi-year drought might begin during or immediately after filling the GERD Reservoir, so it cannot be assumed that Era 2 will precede Era 3. To illustrate the possible effects of such a multi-year drought and the countries' attempts to manage it, we simulate the consequences of the 1972–1987 series of low flows. We first consider how water stored in the HAD and GERD Reservoirs could be used to reduce deficits arising from low water availability. Then we discuss water supplies during a post-drought recovery phase.

*The use of water stored in the HAD Reservoir and the GERD Reservoir as a drought begins*. At the beginning of a multi-year drought, the Eastern Nile riparians should have water stored in both the GERD and HAD Reservoirs, which can be used to mitigate shortages. During a drought, the GERD Reservoir would be drawn down in an effort to continue generating hydropower. Any water released in excess of that which is required for power generation would come at a cost to Ethiopia by foregoing the generation of power in the future. Ethiopia could release water downstream until the GERD Reservoir reached the minimum operating level of 595 masl (or 565 masl if only low-level turbines remained in use).

The most basic strategy to manage a drought would be to guarantee a minimum annual release from the GERD, or alternatively, a firm power production contract from the GERD. In our analysis, we maintain our Era 2 assumption that Ethiopia would continue to operate the GERD to deliver 1600 MW of power whenever possible, Sudan would operate reservoirs to meet its own irrigation and energy generation needs, and Egypt would invoke its current DMP as necessary. As the multi-year drought of the 1970s and 1980s begins, storage in both the HAD and GERD Reservoirs falls quickly (Fig. 7 and Supplementary Table 6). During the drought onset, storage in the HAD Reservoir, however, remains higher than it would have been if the GERD had not been built, causing decreased water deficits to Egypt and increased water availability. Because the GERD releases (for hydropower production) help to boost Nile flows, Egypt experiences four years of reduced shortages until both reservoirs are near their minimum operating level. Up until the peak of the drought (1987), the GERD has *decreased* the cumulative shortages to Egypt from 42 bcm to 27 bcm, reducing shortages by 15 bcm. As shown in Fig. 7, however, the storage in both the HAD Reservoir and GERD Reservoir can become nearly or fully depleted.

*The refilling of the HAD Reservoir and the GERD Reservoir when the drought ends*. At some point, the multi-year drought will end, and a series of average and high floods will arrive. The questions of how fast and in what sequence the HAD Reservoir and the GERD Reservoir should be refilled will be critical and contentious. This case is similar to the initial period of filling the GERD Reservoir but will be much harder to manage because both

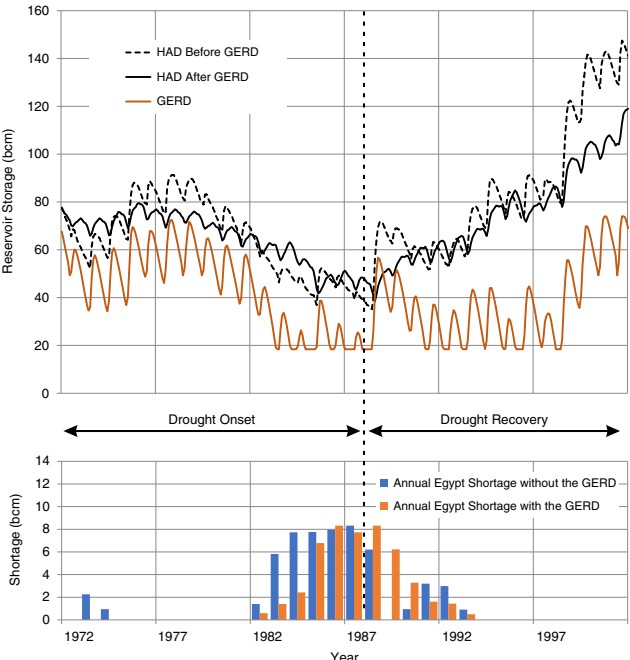

**Fig. 7 Entering drought conditions from *new normal* operations under a historically dry 20-year sequence.** Storage volumes for the GERD and HAD are shown above and shortages to Egyptian planned releases are shown below.

reservoirs will now be empty or nearly empty. If the GERD Reservoir and HAD Reservoir were located in one country, reservoir managers would probably fill the GERD Reservoir first and the HAD Reservoir later subject to meeting most or all downstream water needs[33]. In this transboundary context, however, filling the GERD reservoir first is likely to cause great concern, unless firm guarantees are in place that water will be released to meet the basic requirements of the downstream riparians.

Figure 7 above (and Supplementary Table 6) presents a drought recovery using the 1988–2001 flow series and illustrates the challenge of coming out of such a multi-year drought. We assume that the objective of GERD operators would continue to be the generation of 1600 MW whenever possible. This is a preferred situation for Egypt and Sudan because more water would immediately be released downstream. In this case, the GERD Reservoir is unable to refill quickly and continues to operate close to its minimum operation level (595 masl) while the storage in the HAD Reservoir begins to recover. The GERD Reservoir only begins to refill in years when its inflows reach levels that are high enough to allow generation of the 1600 MW energy target despite the reduced hydraulic head behind the dam.

During this recovery period, an additional 21 bcm of cumulative shortages occur in Egypt when the GERD is in place, compared to 14 bcm of cumulative shortage that would have occurred over the same period without the GERD. The results again demonstrate the challenge of simultaneous refilling of these two large reservoirs. Egypt would be adversely affected during this recovery period, but a decision by Ethiopia to promptly generate electricity would mitigate Egypt's losses. In contrast, a decision by Ethiopia to prioritize refilling the GERD Reservoir by reducing power production would exacerbate Egypt's losses.

## Discussion
Despite the fact that the GERD Reservoir is likely to fill (i.e. Era 1) with minimal reductions in water released for Egypt as determined

by the HAD DMP, the results of the simulations in Figs. 3 through Fig. 5 help to explain why perceptions of upstream and downstream riparians about the risks of filling the GERD Reservoir have diverged. Falling water levels in the HAD Reservoir could be perceived very negatively, and these perceptions might be especially difficult to manage in Egypt where they could be incorrectly interpreted as stemming solely from the filling of the GERD. For example, the GERD may continue to release an agreed annual volume, but droughts may result in below-average contributions from tributaries downstream of the GERD. Indeed, as inflows to the HAD Reservoir are temporarily reduced, falling HAD Reservoir levels will be visible in photographs and satellite imagery. It may be difficult for civil society and governments to fully understand the reasons why these inflows to the HAD Reservoir are so low. It may be known, for example, that rains in Ethiopia are below normal, but mistrust may spread concerning the filling of the GERD and whether agreed upon filling rules are being followed. Similarly, it will be difficult to discern precisely how much water is being withdrawn for irrigation and other upstream uses in Sudan and Ethiopia. Egyptians may assume that this large project has resulted in a loss of water security. Such concerns might spread, and could possibly be exaggerated or distorted by opinions aired through traditional and new media platforms[34,35].

We can also speculate that after the construction and filling of the GERD is complete, a *new normal* period (i.e. Era 2) with a relatively average sequence of Nile flows and without a long multi-year drought could lead to complacency, with neutral to positive outcomes for each riparian. Even in the absence of coordination, Ethiopia will be able to operate the GERD to maximize hydropower generation. As long as the inflows remain somewhere near the long-term average, Egypt will not experience significant deficits, and may even benefit from the buffering effects of the GERD. In this era, the construction of the GERD will be seen as a win-win result for the Nile riparians, which has been Ethiopia's stated position. Egypt may become concerned about lower storage levels in the HAD Reservoir, but would likely become less anxious over time. Sudan will receive multiple benefits due to the upstream flow regulation.

During this Era 2 of the *new normal*, the benefits that Sudan and Ethiopia will receive from the construction of the GERD will far exceed any small losses - chiefly in reduced hydropower production from the HAD – experienced by Egypt[16,36]. However, this *new normal* may reduce the perceived need to coordinate the operations of the HAD and the GERD. This would be unfortunate because the hydrology of the Nile dictates that the riparians must plan carefully for severe multi-year drought.

Our results also demonstrate that a multi-year drought (i.e. Era 3) could have severe impacts that induce difficult water allocation choices. Policy makers already know that behavioral responses to a multi-year drought will be negative and difficult to manage, with challenges in each country exacerbated by the very visual signal of dropping reservoir levels. Even with clearly understood facts, e.g., about reduced rainfall upstream, it will be difficult for Egyptian citizens to understand the full set of reasons for low levels in the HAD Reservoir.

One of the key findings of behavioral economics is that people feel losses much more acutely than gains of comparable size[37]. Moreover, people feel water losses more acutely than they do losses of almost any other commodity[38]. In severe drought conditions, feelings about the loss of water security may become especially acute. We speculate that this could lead to a water panic among irrigators and civil society in Egypt. Fears of a loss of access to water could spread rapidly through the population—increasingly via social media. An analogy may be drawn between a water panic and a financial panic, in which hoarding of seemingly scarce resources (e.g. cash, fuel, food supplies) ensues. Both a water panic and a financial panic are difficult to predict, can spread rapidly, and will be hard to manage without access to data and reassurance by trusted leaders. In a water panic, the state will have to convince the public that essential water supplies are secure.

This reinforces the need for cooperation and agreement, together with a basin-wide, data-sharing platform for coordinated planning, transparent information exchange, and building trust. Cooperative Eastern Nile preparation of contingency plans is best undertaken before these plans are needed, and not in response to a crisis. Drought management plans need to have widespread support before they are implemented so that stakeholders know what will happen in an emergency. During the implementation of such measures, policy makers will need to actively engage with the press and social media to correct misperceptions as they arise and reassure the public that the management plan will be effective and fair. Water uses with a low economic value may be targeted for reductions in supply, and may need to be compensated for their financial losses. The political costs of such curtailments may be significant, but so will be the costs of inaction.

Some economists have argued that Egypt can manage significant reductions in releases from the HAD Reservoir without large reductions in economic output[19]. This argument holds that targeted reductions in water supply to low-value water-intensive crops would allow Egypt to pull through a multi-year drought with minimal economic consequences. But just as macro-economic models are not able to account for the human emotions underpinning a financial panic, these economic models do not account for the possibility of a water panic. Feelings may run high if people feel that they are unjustly denied access to water or if the burden falls disproportionately on poor, vulnerable farmers.

In summary, sharing the scarce waters of the Nile Basin involves balancing competing objectives under conditions of uncertainty. In this paper, we synthesized modeling results into narratives based on three eras that help to explain the implications of the GERD on the Nile system. We believe that this framing will help both technical and nontechnical audiences better understand the dynamic situation in this international river basin, and deepen their collective understanding about several key behavioral features of this complex system. Moreover, we use these narratives to highlight the need for careful management of the built infrastructure in the Nile, to consider how potential outcomes might be perceived by civil society and decision-makers, and to identify the circumstances under which these perceptions of risk could lead to a water panic. The analysis serves to highlight the challenges faced in the ongoing and future negotiations among Eastern Nile riparians. Recognizing the origins and implications of these challenges may assist movement towards a resolution.

In providing such reflections, we have sought to provide an accessible description of the context of the Nile Basin system, and to show how completion of the GERD will influence its hydrological behavior. Our simulations illustrate how the construction of the GERD will affect the three Eastern Nile riparians and help to highlight several critical situations that need to be considered in negotiations related to GERD and HAD operations. The first critical situation is the imminent period of filling of the GERD Reservoir (Era 1). We recognize that the three countries have been actively engaged in negotiations to reach a resolution over this pressing issue. The necessary management decisions relate to whether there will be agreed releases from the GERD and whether a more sophisticated strategy should be adopted to manage an extraordinary, yet very plausible, severe drought during the initial filling.

Once the GERD Reservoir is full, in a sequence of years of average or above average Nile flows (Era 2), Egypt is unlikely to need to reduce water releases from the HAD, provided that upstream irrigation withdrawals have not significantly increased. However, the HAD Reservoir will usually operate at lower levels than at present. Sudan will benefit from steadier Nile flows, including increased summer flows and reduced floods and sedimentation. Ethiopia will benefit from the generation of 16 TWh per year of hydropower.

However, a severe multi-year drought (Era 3) is inevitable at some point in the future. This will be a critical event in terms of managing water risks in the Eastern Nile. In advance of such a drought, a comprehensive basin-wide management plan needs to be agreed upon, including a management policy for the GERD. Such agreements should specify how the reduced flow of the Nile will be shared when storage is depleted in both reservoirs, and will need to balance power generation and consumptive use. At least as challenging will be the issue of how quickly and in what sequence the HAD and GERD Reservoirs should be refilled. To maintain confidence that proper planning has taken place, coordinated communications with the media and public in all riparian countries will be important.

Reaching an agreement on a filling strategy is important now. However, it is also urgent to begin planning for the inevitable severe multi-year drought that will occur in the future. No one can predict when such an event will occur, but we can anticipate both its implications for system outcomes, and how it will be perceived by different stakeholders. It is in the interest of the Nile riparians, as well as the global community, for agreements to be in place to prevent a water panic from developing. Basin-wide drought planning can occur concurrently or begin immediately after an agreement over filling is reached.

Developing robust contingency plans is not an insurmountable task. In most years the GERD and HAD will require only modest coordination, and data transparency should be sufficient for effective planning. Yet this analysis demonstrates that nobody should be under the illusion that unilateral decision-making will allow management of a severe multi-year drought. It is incumbent on decision makers to reach agreement on the mechanisms and strategies needed for managing a new set of water-related risks.

## Methods

**Analysis framework and key assumptions**. This study uses a detailed analytical water resources systems model called the Eastern Nile RiverWare Model (ENRM)[20] that simulates the current and future management of the river from 2020 until 2060. This simulation tool was developed using the rule-based RiverWare platform[39], which has been used extensively by water managers and researchers to evaluate complex river and reservoir management operations for river basins around the world. Operating rules for each reservoir are translated into a set of prioritized logical statements that specify the reservoir releases required to meet a set of objectives for each water management infrastructure. Depending on the specific purposes of the infrastructure, the objectives for the Nile reservoirs may include satisfying agricultural and municipal needs, meeting power generation demands, achieving seasonal target elevations for sediment transportation, guaranteeing minimum monthly flow requirements, and implementing flood management and shortage avoidance policies. The key modeling assumptions are described below and in the Supplementary Information, and the configuration of the ENRM is described in previous studies[20,21].

**Modeling the filling period (Era 1)**. The filling policy used to demonstrate the range of outcomes is based on a preliminary proposal of the official National Independent Scientific Research Group (NISRG) that convened from 2018 to 2019. The NISRG was comprised of researchers appointed from each of the three countries. This policy assumes that the GERD Reservoir will be filled to required minimum turbine operation levels during the first two years of filling. Specifically, in year one, 4.9 bcm will be retained to bring the water level to the operation level for two low-head turbines—565 m above sea level (masl) (Supplementary Fig. 2). All remaining flows will pass through the dam's temporary "low block" spillway since hydraulic capacity of the outlets are limited[27]. In the second year, a larger

amount of water will be retained (13.5 bcm), to reach 595 masl and allow testing of additional turbines. These flows represent 5% and 15% of the natural average annual inflow to the HAD Reservoir in years 1 and 2, respectively, and 10% and 28% of average flow into the GERD Reservoir. From that point onwards, we assume that Ethiopia would release at least 35 bcm each year for the remainder of the filling period (to fill the remaining 55.6 bcm between 595 masl and the full supply level (FSL) of 640 masl). Releases would be made as evenly as possible throughout the water year (July to the following June). Therefore, water would be retained in the GERD Reservoir during the peak months of the Blue Nile flood (July, August, and September) and released throughout the year. Although the filling process may officially terminate when the reservoir elevation reaches 625 masl during the dry season, Ethiopia would continue to fill the GERD Reservoir up to the FSL of 640 masl at the peak of the flood season to maximize hydraulic head and energy generation.

During filling, we assume that Egypt will continue to attempt to release 55.5 bcm annually from the HAD Reservoir (4 bcm pumped directly from the reservoir to the Toshka project and the remainder through the dam outlet works) unless storage levels fall below volumes specified in the current Drought Management Policy. This policy specifies reductions of releases of 5%, 10%, and 15% if the storage levels in the HAD fall below 60, 55, and 50 bcm, respectively. These reductions are first reduced by reducing or eliminating the Toshka pumping demands, then HAD releases are further reduced if needed to meet the DMP reduction target (see Supplementary Fig. 1). We assume that Sudan will withdraw the current estimated total of 16.7 bcm from various diversion locations and that evaporative losses from Sudanese reservoirs vary based on reservoir levels[21,40]. A sensitivity analysis of this assumption is provided in Supplementary Note 6 and Supplementary Fig. 7. Ethiopia is assumed to withdraw 0.45 bcm each year for the Finchaa irrigation scheme[41]. Historical irrigation sites around Lake Tana are not explicitly modeled, but inflows to the lake are derived via calibration with the outflows of Lake Tana[3], hence they are implicitly assumed. Consumptive use values during filling are estimated and assumed to not expand into the immediate future. Assumed water usage in the model does not reflect any endorsement of water rights.

**After filling (Eras 2 and 3)**. Once the GERD FSL is reached, we assume a regular hydropower production of 1600 MW based on an estimated 90% maximum reliable hydropower generation rate[21]. Ethiopia's electricity demands are projected to increase from 11.1 TWh in 2020 to 27.3 TWh in 2030[42], suggesting that increasing their current 4400 MW of installed generation capacity with the 5150 MW from the GERD will soon be justified. We also assume the GERD pool elevation will be lowered to a maximum of 625 masl in June of each year for seasonal flood planning. We assume Egypt will continue to try to release 55.5 bcm each year from the GERD Reservoir, subject to the DMP described above. Although the GERD could allow Sudan to increase its irrigation withdrawals to the full allocation under the 1959 Nile Water Agreement of 18.5 bcm as measured at Aswan, the amount of additional water that Sudan can extract is uncertain due to ungauged diversions, a lack of transparent reporting, debates over how flood waters accumulated during previous years should be accounted for, and uncertainties over how evaporation from various reservoirs should be considered. For our analysis, we assume a continued withdrawl of 16.7 bcm from the current points of diversion, exclusive of reservoir evaporation losses. Egypt and Sudan disagree as to how evaporation losses from Sudanese reservoirs are to be counted in the calculation of Sudan's allocation of 18.5 bcm in the 1959 Nile Waters Agreement. Egypt's interpretation of the Agreement is that evaporation losses from the construction of Merowe, the heightening of Roseires Dam, and any new Sudanese dams should be included as part of Sudan's allocation of 18.5 bcm. Sudan argues that the calculation of the Nile flow at Aswan (84 bcm) already considers the evaporation and seepage losses which occur inside Sudan, including evaporation losses from current and future dams. Thus, Sudan's position is that evaporation losses from reservoirs constructed after the 1959 agreement should not be counted as part of its 18.5 bcm allocation.

Future Ethiopian developments relevant for Eras 2 and 3 are assumed to withdraw an additional 1.0 bcm from medium-term development plans in the Upper Beles (including transfers to the Dinder River), Anger and Arjo tributaries[10], bringing the total Ethiopian diversion to 1.45 bcm—over and above the amounts used for irrigation around Lake Tana. Assumed future withdrawals are considered country-level targets and again do not reflect any endorsement of water rights.

**Hydrologic conditions**. Selecting traces to use in the three eras was based on an analysis of naturalized inflows to the HAD Reservoir from 1900 to 2018[3,4]. This naturalized data set reconstructs conditions prior to major human influence by removing the effects of upstream agricultural depletions and management. The period from 1900–2002 used a hydrologic reconstruction derived from 41 gauge sites by the consulting firm Deltares for the Nile Basin Initiative[3]. The naturalized flows at Aswan from 2003 to 2018 were derived using the gaged flows at Dongola[4] and adjusting for estimated Sudanese uses during this period increasing from 14.6 bcm (2005) to 16.7 bcm (2017). From this data set, we select representative historical periods that describe average, high and low flow conditions in the hydrologic record, as shown in Fig. 2 and Supplementary Note 5 and Supplementary Table 3.

We recognize that future conditions will not replicate those of the past and that more severe conditions could materialize, especially with a changing climate[43]. The representative historical flow sequences nonetheless allow us to illustrate a wide range of hydrological events and management responses. Supplementary Fig. 5 demonstrates the range of 10-year average flows and the traces selected for the GERD Reservoir filling period (Era 1) and a drought recovery period (end of Era 3). Supplementary Figure 6 demonstrates the range of 20-year average flows and the traces selected for the *new normal* (Era 2) and the period entering a severe multi-year drought (start of Era 3).

From the perspective of a hydrologist planning for future climatic events, the use of historical flows to simulate a water resource system is restrictive because it ignores floods, droughts, and sequences of flows that are possible but not within the historical record[44,45]. Advanced stochastic methods for synthetic hydrology generation have been available for decades and continue to evolve[46] and the stochastic character of Nile flows has been extensively studied[47–49] resulting in state-of-the-art synthetic generation techniques that demonstrate a very large range of future possibilities for the Nile[50–52]. However, the Nile debate within the current and formal negotiations has focused primarily on what would happen if the notorious drought of the 1980s were to recur. More generally, the use of observed historical flow series remains standard practice in many complex transboundary river basin negotiations[53,54].

Water resources planners and analysts have never adequately explained why the use of historical flow series is still the predominant method used in high-level river basin negotiations, despite the widespread recognition that the climate has changed and past hydrological records are unlikely to be a good guide to the future[55]. The use of historical flows has three advantages that have not been fully appreciated. First, simulations using historical flows are more easily understood and transparent, because they relate to real lived experience. Second, they are perceived to be less susceptible to bias and manipulation by modelers[56–58]. This is especially an advantage in river basins like the Nile with widely divergent climate change forecasts[32,59–61].

Third, and most importantly, historical traces can be more readily used to create a convincing 'policy narrative' or 'story'. The advantages of storytelling as an analytical approach for both describing status quo conditions and illustrating the consequences of different policy interventions are now widely understood in domains of business[62] and policy analysis[63]. However, many water resources professionals have not fully appreciated the importance of creating such a policy narrative to better understand and communicate the characteristics of a complex multi-faceted water resources problem. Historical flows have happened in the past and they often resonate emotionally in people's memories. The storyline associated with historical hydrological traces is simple to explain: what would be the consequences if these flows occurred again. Importantly, such narratives are not just a means to communicate to nontechnical audiences; they are also valuable for modelers to better understand and place their results in a historical context.

## Data availability

The data that support the findings of this study are available on request from the corresponding author [D.W.]. The data are not publicly available due to state restrictions and contain information that could compromise research participant privacy/consent.

## Code availability

The model that support the findings of this study are available on request from the corresponding author [D.W.].

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

## Acknowledgements
We would like to thank the following individuals for their comments and suggestions on previous drafts of this paper—Mohammed Alam, J. Anthony Allan, Mohammed Basheer, Thinus Basson, Alan Bates, Hesham Bekhit, Donald Blackmore, Ana Elisa Cascao, Simon Dadson, Ariel Dinar, Jerome Delli Priscoli, Asim I. El-Maghraby, Alaa Elzawahry, David Grey, Giorgio Guariso, Julien Harou, Kingsley Haynes, Daniel P. Loucks, Yasir Abbas Mohamed, Scott Moore, Claudia Sadoff, Abdulkarim H. Seid, Kenneth Strzepek, Eelco Van Beek, John Waterbury, Xun Wu, and Yasmin Zaerpoor.

We also appreciate the financial support for this research provided by the Oxford Martin Programme on Transboundary Resource Management.

## Author contributions
K.G.W. and E.Z. developed the water management model described in the study; D.W. and M.J. led the conceptual development of eras; D.W., K.G.W., and M.J. led the statistical analysis and development of the hydrologic scenarios, J.W.H., D.W., and E.Z. contributed to the interpretations of societal implications and risks, all authors contributed to the writing of the article.

## Competing interests
K.G.W. has provided consulting services related to model development, Nile water management, and understanding the implications of the Grand Ethiopian Renaissance Dam since 2012 for the Nile Basin Initiative, the World Bank, Stockholm International Water Institute and the Water Resources Research Institute, Egypt. Contributions to research include GIZ on behalf of the German Federal Foreign Office. He has had ongoing academic collaborations with University of Khartoum, University of Addis Ababa, and Cairo University since 2012. M.J. currently provides consulting services (beginning in 2018) to the Nile Basin Initiative (NBI) to assist with the development of a hydroeconomic model to analyze water infrastructure investments in the Nile Basin. During 2017 he also provided technical support on economic analysis through the NBI's Economist Forum. From 2006 to 2009 he was a member of the Eastern Nile Scoping Study Team, whose work was commissioned by the Eastern Nile Council of Ministers and funded by the World Bank. D.W. served as Chair of the Board of the Environment for Development Initiative until 2019, which has a research center in Addis Ababa that works on water resource management issues in Ethiopia. From 2006 to 2009 he was a member of the Eastern Nile Scoping Study Team, whose work was commissioned by the Eastern Nile Council of Ministers and funded by the World Bank. In 2017–2018 he served as a member of an external technical advisory committee for Deltares on a project —financed by the Nile Water Sector, Government of Egypt—to study the effects of the Grand Ethiopian Renaissance Dam on the Nile system. E.Z. provided consulting services to the Nile Basin Initiative (NBI) from 2007 to 2012, providing technical expertise to the development of the Nile Basin Decision Support System, and also in 2015–2017, providing assistance to NBI's Strategic Water Resources analysis, a collaborative work with stakeholders from the Nile Basin countries. J.W.H. has no competing interests to declare.
