## [Peer Review File · Nature Communications]

REVIEWER COMMENTS

Reviewer #1 (Remarks to the Author):

This is a good paper and does not need changes

Reviewer #2 (Remarks to the Author):

This is an excellent analysis, it has a simple approach to a complex multi-faceted problem and is very clearly written. The subject and results are highly topical and policy relevant – its simplicity is a major strength (not least because the results are important for a diverse audience, many of whom maybe non-technical) but in a research sense it is rather a drawback - the work is great, very thorough and methodologically and analytically rigorous (the assumptions made are clearly articulated and generally reasonable) and with interesting results but I'm unsure there is enough research advancement here to warrant publication as a research article in Nature Communications. This is to some extent an editorial decision but at the very least I'd ask the authors to explain more clearly (in the response and in the paper) what the main research advances are and how they are significant in relevant fields. The approach is not very novel, the modelling uses an established operational modelling system with no modifications, and the analysis while excellent is not really innovative so I'm not sure this is really a research paper, it is more of an important report, and as such might be better suited as a Perspective/Policy forum type piece?

Some questions about where the innovation might lie.....

There is reference to a cluster of recent citations on the GERD (lines 71 – 73) – how does this paper improve/add to these, are the results/insights different? (to what extent do the papers that generate synthetic inflow series produce results outside the observational series?) Is there broader potential to use this simple analytical approach in other cases? (what are its main strengths and weaknesses?)

The linking between hydrological analysis and policy implications is interesting and very clear (the discussion of potential government and civil society perceptions of the risks is strong) but are there effective precedents for drought management plans that could be referred to?

To what extent are standard design practices already doing this type of analysis and are there precedents for other international basins? How does this approach advance standard engineering design for infrastructure to manage variability?

Is there anything about the variability in Nile flows that makes this situation different or comparable to other multi-reservoir problems....?

Lines 484-485 - 'This "historical trace" approach is restrictive relative to previous papers that have employed large numbers of stochastic flow sequences^{18,19}, but results are more intuitive and easier to interpret' – I agree with this but can you make a stronger case for why this is so – the simple set-up is very helpful - why?

If the authors can provide a convincing argument for research significance and weave this into the paper then I think it would be suitable for publication as a research paper – the quality of the work is excellent and I don't see any other specific requirements for revision. Below are some thoughts that could be considered but none are essential.

Some points for consideration;

Recent work in Sheen et al. (2018) has shown some skill in multi-year rainfall hindcasts using a decadal prediction system that gives significant correlations with observed rainfall averaged over a 2-5 year period across the Sahel, including some parts of the Blue Nile basin and central Sudan (see their figure 1). Advance warning of the next few years of rainfall conditions could inform early steps towards drought management in cases where dry conditions are forecast – I was recently involved in some scoping work on possible applications of multi-year forecasting in Sudan and (noting major scientific and operational challenges) there was considerable interest in the potential of multi-year (and extended lead time seasonal) forecasts for improved hydropower/irrigation management in the Blue and Main Nile systems (Ward and Conway, 2020).

How realistic is a demand level for GERD to generate electricity of 1,600 MW? (is there any recent information about the current supply load and how soon new grid infrastructure might be in place?)

It's a shame the authors don't have flow data more recent than 2002 (Figure 2), 18 years ago.... (any indication of how this missing period compares with the selected 'traces'?). More work on assessing the realism of publicly available near real-time satellite-derived rainfall estimates (e.g. CHIRPS) over the contributing basins might help with transparency for riparians and could act as a proxy for recent flows (good rainfall - runoff relationships exist for the Blue Nile at basin-scale, and there is near real-time satellite record for the level of Lake Tana that could be helpful for extending lead times in early warning decision triggers).

There is a lot of work looking at implications of multi-year variability in the Nile system (not least Hurst's pioneering work on the Nile) – the Equatorial Nile and Eastern Nile systems are not strongly correlated.... 'traces' or stochastic models should ideally treat them separately. What would happen if a sequence of low flow years in both systems occurred (I think White Nile flows were well above long-term average during 1978-87 and buffered the low Eastern Nile flows), or wet as was possibly the case pre-1900?

I think it's fair to say there is large uncertainty about short and long term flow patterns that result from multi-year drivers of variability and climate change. However, warming will continue and the already substantial evaporation losses from the reservoirs will potentially increase....is this worth a mention (cite estimates of changes in evaporative losses from the reservoirs)?

The dry period used as a trace in this paper prompted Egypt to develop capacity for modelling and planning, a department focussing on monitoring and forecasting of Nile flows and a strong water policy focus on scarcity, but the subsequent (prolonged) return to higher flows relaxed the level of concern (Abu Zeid and Abdel-Dayem, 1992; Conway, 2005) – the GERD could act as another trigger for capacity building for DRM (and potentially cooperation) - this analysis is a timely reminder that a multi-year sequence of below average flows is a very real threat.

References

Abu-Zeid, M., Abdel-Dayem, S., 1992. Egypt's programmes and policy options for facing the low Nile flows. In: Abu-Zeid, M.A., Biswas, A.K. (Eds.), *Climatic Fluctuations and Water Management*. Butterworths and Heinemann, Oxford, pp. 48–58.

Conway, D. (2005). From headwater tributaries to international river: Observing and adapting to climate variability and change in the Nile basin. *Global Environmental Change*, 15(2), 99-114.

Hurst, H.E., Black, R.P., Simaika, Y.M., 1965. *Long Term Storage: An experimental Study*. Constable, London.

Sheen, K. L., Smith, D. M., Dunstone, N. J., Eade, R., Rowell, D. P., & Vellinga, M. (2017). Skilful prediction of Sahel summer rainfall on inter-annual and multi-year timescales. *Nature communications*, 8(1), 1-12.

Ward, W.N. and Conway, D. (2020) Applications of interannual-to-decadal climate prediction: an exploratory case study for rainfall in the Sahel region of Africa. *Climate Services* 18 (2020) 100170

Declan Conway

Referee #3

Review of the paper: Understanding and Managing New Risks on the Nile with the Grand Ethiopian Renaissance Dam

25. It is important to extend the average flow of the Nile to cover the period 2002 up to 2019.

34. HAD total design capacity was 162 bcm and dead storage was 31.6 bcm. However, due to reservoir siltation of 7 bcm the capacities are now different, 46% is deposited in the live storage.

47. HAD design capacity was 162 bcm, now 155 bcm. 60. GERD new capacity of turbines is 6450 M and hence annual power generated will be in the range of 15962-17500 Gwh.

141. There is no logic to build the assumptions on HAD storage of 120 bcm. It is well known that the first or initial filling is going to start on July 2020 where HAD water level will be 179.6 m, or a little bit less, with a capacity around 147 bcm. The paper has to work on actual values not hypothetical ones.

149. Fig. 4. Use the actual situation (level and volume of water) expected to be at HAD on the first of July 2020.

456. Sudan total abstraction is 12.2 bcm at Aswan and not 16.7. This is based on actual measured values.

469 – 471. Diversion of Sudan is measured by both Sudan and Egypt authorities and not debated. This is based on the Permanent Joint Technical Commission, PJTC, record.

472. Assuming 16.7 bcm withdrawal by Sudan is not right.

474. Assuming withdrawal of 2 bcm from lower Beles is exaggerated since GERD reservoir is going to submerge 80 – 90 km of the arable land in this part of lower Beles.

547. Roseries reservoir capacity is now 5.9 bcm. 2 709. Table S1. Rosereis minimum pool elevation is now 471 m instead of former level of 467 m. Also review HAD levels.

712. Table S2. Sudan water demand is now 12.2 bcm at Aswan. Ethiopia water demand to be also reviewed based on what is mentioned in 474.

730. Table S4, and Table S5. Review Sudan annual evaporation based on: Merowi 1550 m, Sennar 300 m and Roseries 650 m. Review also HAD evaporation.

Conclusion

1. This paper overlooked that the existing situation in HAD reservoir is a combination of the unutilized share of Sudan quota (6.5 bcm at Aswan) and the extra flood waters accumulated during the last three decades that should also be divided among the two countries (3.5 bcm for each) according to the 1959 Nile Water Agreement.

2. Evaluation of the first filling has to start from the actual level and capacity of HAD on first of July 2020 not based on a hypothetical level and capacity.

3. The paper is timely and covering a pertinent issue now under consideration and negotiation by the three countries.

Prof. Seifeldin Hamad Abdalla, ED Nile Basin Initiative. May 2020. Entebbe, UG

Response to Reviewers' Comments - "Understanding and Managing New Risks on the Nile with the Grand Ethiopian Renaissance Dam"

Reviewer #1:

This is a good paper and does not need changes

Authors' Response: We appreciate Reviewer #1's positive assessment of our paper.

Reviewer #2:

Comment No. 1: This is an excellent analysis, it has a simple approach to a complex multi-faceted problem and is very clearly written. The subject and results are highly topical and policy relevant – its simplicity is a major strength (not least because the results are important for a diverse audience, many of whom maybe non-technical) but in a research sense it is rather a drawback - the work is great, very thorough and methodologically and analytically rigorous (the assumptions made are clearly articulated and generally reasonable) and with interesting results but I'm unsure there is enough research advancement here to warrant publication as a research article in Nature Communications. This is to some extent an editorial decision but at the very least I'd ask the authors to explain more clearly (in the response and in the paper) what the main research advances are and how they are significant in relevant fields. The approach is not very novel, the modelling uses an established operational modelling system with no modifications, and the analysis while excellent is not really innovative so I'm not sure this is really a research paper, it is more of an important report, and as such might be better suited as a Perspective/Policy forum type piece?

Authors' Response: The reviewer raises an important question, i.e., what is the research contribution? The reviewer says that our analysis is “excellent”, “very clearly written”, “highly topical”, “policy relevant”, that “its simplicity is a major strength.” But the implication is that this paper is somehow simply a translation of more complex, sophisticated technical analyses to a nontechnical audience that are already well understood by scientists working in the Nile basin. Thus, the reviewer says that the approach is “not very novel” and implies that its value lies in explaining complex results that scholars already understand.

In the revised manuscript, we address Reviewer #2's suggestion that we “explain more clearly (in the response and in the paper) what the main research advances are and how they are significant in relevant fields.” In fact, we make two substantive contributions. First, we provide a new narrative about the management of risks in the Nile basin that both technical and nontechnical audiences can understand. Our description and framing around the “Three Eras” with different risk management problems is new, and it is not currently understood by either technical or nontechnical audiences in the Nile basin. The narrative combines key aspects of the physical representation of the basin with operations that reflect realistic objectives of the riparians, framed using hydrologic scenarios from the observed record. Once someone reads this narrative, it perhaps “seems simple” and obvious, but we argue that the results are in fact a reflection of a powerful and compelling narrative that reveals complex behavior that has not been previously characterized. Indeed, it is often hard for modelers of dynamic nonlinear systems to understand the implications of their own results. There are approximately a dozen Nile modeling teams around the world trying to better understand the problem of the optimization of the multi-reservoir system and risk management in the Nile basin, and these teams have

published many papers on the problem including ones specifically about the GERD. However, this is the first paper to use a narrative methodology to illustrate the essence of the new situation that the basin is now facing. To clarify this contribution, we now cite – in the methods section – the literatures in business and policy analysis that describe the importance of using narratives to explain both status quo conditions and the consequences of policy interventions that have rarely been applied in the hydrology and water resources domain.

The second major contribution of this paper is to reveal how scenarios of water risks on the Nile may manifest as observable impacts and subsequently propagate by affecting the emotions of policy makers and civil society in the Nile basin. The many modelling teams working on the Nile are aware of these political tensions and are more or less explicit about the conflicting goals of the Nile riparians. However, because their analysis has largely been in terms of metrics like hydropower production and economic impact, they have not engaged with the potential political and societal reactions to the observable impacts of construction of the GERD – namely perceptions of changed water levels elsewhere in the system. Until our paper, no one had discussed the linkage between the results of the Nile modeling efforts and how these consequences are likely to induce emotions like panic and fear. For example, we show that during the filling of the GERD, the HAD reservoir will fall for several years as a physical consequence of withholding more water in Ethiopia, even during moderate hydrologic conditions. We then demonstrate that the levels in the HAD reservoir are likely refill after the GERD reaches its operational level. Unlike other modeling efforts, we discuss how this falling level of the HAD may induce fear and panic in Egypt, even though the HAD Reservoir will eventually recover. We argue that these perceptions need to be understood and demand careful attention from policy makers lest they lead to knock-on effects that induce even more tension in the basin.

Comment No. 2: Some questions about where the innovation might lie.....

There is reference to a cluster of recent citations on the GERD (lines 71 – 73) – how does this paper improve/add to these, are the results/insights different? (to what extent do the papers that generate synthetic inflow series produce results outside the observational series?) Is there broader potential to use this simple analytical approach in other cases? (what are its main strengths and weaknesses?)

Authors' Response:

In the revised manuscript, we address Reviewer #2's comments on the relation of our work to the existing literature by adding a new section in the Supplemental Materials ("Relation to the Existing Nile Systems Analysis Literature"). We explicitly position this paper within the array of previous studies. The new text reads:

"The existing systems analysis literature on the Nile focuses largely on methodological issues in the simulation and optimization of multi-reservoir systems. Some papers include stochastic flow sequences¹⁻⁴ and various climate scenarios⁵⁻¹⁰, but most authors use simplified representations of the Nile infrastructure in order to illustrate new methods. Several studies have sought to quantify the value of consequences in economic terms^{1,5,10-13}. Although such quantification of economic consequences is useful when applying formal decision analysis, most presume some degree of idealized or optimal operations. In actual negotiations on contested rivers, decision makers are typically more interested in directly observable consequences of proposed agreements and plans (e.g., reservoir releases and levels).

We have been a part of and learned from this systems literature on the Nile. As a result, in this paper we distill the salient aspects of the system behavior and reflect these aspects in a small number of selected flow sequences. In contrast to the existing literature, the analysis in this paper focuses on the actual operational characteristics and observable consequences (e.g. reservoirs releases and levels) of the Eastern Nile system, which we believe will make our results more credible to decision makers. Although most of the papers in this literature describe the transboundary context and acknowledge the differing objectives of the various Nile riparians, none uses a narrative methodology to assist decision-makers' interpretation of modeling results.

We have used historic flow series as part of a narrative methodology to complement rather than supersede stochastic simulations and associated uncertainty methodologies that involve more exhaustive or probabilistic analysis of extreme events^{4,14}. These stochastic methods are attractive for exploring the inevitable variability and uncertainty in river flows, and have been found to lead to more efficient water resource management plans¹⁵. However, we question their applicability in highly contested situations where actors have conflicting understandings and where trust is low. The appropriate use of combinations of methods will depend on context, but we emphasize that the use of historical, narrative methodologies such as presented in this paper should not be viewed as a methodologically inferior approach."

In the revised manuscript, we also address Reviewer No. 2's two questions ...

- 1) ***Is there broader potential to use this simple analytical approach in other cases? (what are its main strengths and weaknesses?); and***
- 2) ***The linking between hydrological analysis and policy implications is interesting and very clear (the discussion of potential government and civil society perceptions of the risks is strong) but are there effective precedents for drought management plans that could be referred to?***

We now argue more clearly that narrative methodology coupled with historical flow series can be more widely used as a complement, not a substitute for systems techniques using stochastic methods.

Regarding the question ***"to what extent do the papers that generate synthetic inflow series produce results outside the observational series?"***

Yes, in our revised text we have acknowledged that this is possible:

"From the perspective of a hydrologist planning for future climatic events, the use of historical flows to simulate a water resource system is restrictive because it ignores floods, droughts and sequences of flows that are possible but not within the historical record^{16,17}. Advanced stochastic methods for synthetic hydrology generation have been available for decades and continue to evolve¹⁸ and the stochastic character of Nile flows has been extensively studied¹⁹⁻²¹ resulting in state-of-the-art synthetic generation techniques that demonstrate a very large range of future possibilities for the Nile²²⁻²⁴."

Nonetheless, we do not believe that more extensive sampling of synthetic flow conditions would substantively change our results. A thought experiment of the possible implications is as follows:

- Era 1: Drier conditions would prolong the time required to fill the reservoir (effecting Ethiopia's objectives), and depending on the release arrangement agreed, increase the impact on Egypt.
- Era 2: This is by definition a 'central' sequence within the observed range
- Era 3: We have analyzed the impact of a historic multi-year drought. The implications would only be amplified for more prolonged or severe droughts, but the underlying challenges and perceptions would be the same.

We do not think our narrative would change substantively in more severe conditions. Synthetic sequences are more useful for quantification, impact analysis and risk analysis, but selected traces are sufficient, and arguably more effective to develop a convincing narrative, especially given that the historical record offers circumstances (in eras 1 and 3) that illustrate substantial strain on the system.

Reviewer 2's Question: The linking between hydrological analysis and policy implications is interesting and very clear (the discussion of potential government and civil society perceptions of the risks is strong) but are there effective precedents for drought management plans that could be referred to?

The development or modification of policy, especially drought management plans, in response to hydrologic variability is common in water scarce regions such as the western U.S. The Colorado River Basin (CRB) is similar to the Nile in having two large multi-year storage reservoirs that operate to supply water to seven states and Mexico. The 2019 Drought Contingency Plan²⁵ (Congressional Bill 2030, 2019), a plan to reduce the risk of reservoirs declining to critical elevations, was designed to address the current 20-year drought of record. Hydrologic analysis played a major role in the negotiations. The Bureau of Reclamation employs a wide range of methods for projecting future hydrology in the CRB including resampling of the 112—year historic record, ~700 years of paleo data (from tree ring reconstructed hydrologies, and GCM generated climate change projections. Related to our historic narratives, the CRB also considers the 1988-2018 “Stress Test” observed period that removes earlier high flow periods and includes the recent drought of record.

Comment No. 3: To what extent are standard design practices already doing this type of analysis and are there precedents for other international basins? How does this approach advance standard engineering design for infrastructure to manage variability?

Authors' Response: We make clear that the use of selective historical traces is a standard design practice. What is not common is the organization of the results of historical traces into a narrative for both technical and nontechnical audiences. We now reference the policy analysis literature on the importance of using stories to persuade people and policy makers of both a problem description and an appropriate policy intervention.

In the revised manuscript, we expand our argument that the selective use of historical flow series to better visualize the likely consequences of different hydrological sequences—and organizing these historical traces into an easy-to-understand narrative- is an important contribution and is likely replicable in other river basins.

We wish to emphasize that the challenges that our paper addresses is not one of engineering design. The GERD has already been designed, so the question is how to operate a completed dam in a way that does not intolerably harm the interests of downstream riparians. This is a different question to the design question, but it would preferably be dealt with at the same time as engineering design. In other words, design and operational rules would typically be dealt with at the same time in a less politically-divisive context. Best engineering practice would use a combination of historical traces and stochastic series. Leaving the agreement on operational questions until the reservoir is filling is quite unusual, so one could argue that good engineering and water management practice has already been violated. But we can't turn the clock back, so our methodology is intended to shed new light on an extremely important and very challenging situation. Though this is an unusual situation, it is unlikely to be unique given the number of dams on transboundary rivers that are planned, and the past evidence of tensions

over such projects in other contexts (e.g., the Tigris/Euphrates, the Mekong, the Indus) where the clarity of the narrative is more acutely critical than the breadth of the stochastic possibilities.

Comment No. 4: Is there anything about the variability in Nile flows that makes this situation different or comparable to other multi-reservoir problems....?

Authors' Response: After the completion of the GERD, the Nile will be unique in one important respect. It will be the only international river with two large over-year storage reservoirs in two different countries in a water-scarce basin. However the Colorado River Basin (CRB), the most similar international water-scarce basin, has two over-year storage reservoirs that are "owned" by two groups – the Upper Basin States, and the Lower Basin States, and has highly variable flows, comparable to the Nile. These two groups, along with Mexico as the 'third' major user, represent very different water use perspectives. Negotiating the operating policy of the river has historically depended on compacts, laws, court rulings and international treaties - all time-consuming, bureaucratic and contentious processes. The new 21st century paradigm is collaborative negotiations with shared modeling and analysis, and has resulted in positive outcomes including the 2007 Interim Guidelines for Lower Basin Shortages and Coordinated Operations of Lake Mead and Lake Powell. The Eastern Nile countries could benefit from a similar process, keeping in mind the narratives presented in this paper.

Comment No. 5: Lines 484-485 - 'This "historical trace" approach is restrictive relative to previous papers that have employed large numbers of stochastic flow sequences 18,19, but results are more intuitive and easier to interpret' – I agree with this but can you make a stronger case for why this is so – the simple set-up is very helpful - why?

Authors' Response: In the revised manuscript, as discussed above, we emphasize the importance of using the historical flow series to create a narrative, and we reference the policy analysis and business literature on the value of narrative methods for both describing status quo conditions and illustrating the consequences of policy interventions.

Comment No. 6: If the authors can provide a convincing argument for research significance and weave this into the paper then I think it would be suitable for publication as a research paper – the quality of the work is excellent and I don't see any other specific requirements for revision.

Authors' Response: We appreciate Reviewer #2 suggestion, and in the revised manuscript we have tried to weave a convincing argument for research significance into the paper. In particular, we have

1. Added a new opening passage to the Conclusions that emphasizes the originality of the 'three Eras'
2. Explained, in the Methods section, how our use of historic traces complements other hydrological approaches and is particularly applicable in this context
3. Introduced a new section at the start of the Supplemental Material which positions our research in relation to previous studies of the Nile.

Comments No. 7: Recent work in Sheen et al. (2018) has shown some skill in multi-year rainfall hindcasts using a decadal prediction system that gives significant correlations with observed rainfall averaged over a 2-5 year period across the Sahel, including some parts of the Blue Nile basin and central Sudan (see their figure 1). Advance warning of the next few years of rainfall conditions could inform early steps towards drought management in cases where dry conditions are forecast – I was

recently involved in some scoping work on possible applications of multi-year forecasting in Sudan and (noting major scientific and operational challenges) there was considerable interest in the potential of multi-year (and extended lead time seasonal) forecasts for improved hydropower/irrigation management in the Blue and Main Nile systems (Ward and Conway, 2020).

Authors' Response: We agree that forecasting systems are demonstrating increasing skill to an extent that forecasts could be worth using for enhanced reservoir operation. We are, however, reluctant to introduce the possibility of using forecasts into our paper, for a number of reasons:

1. We have endeavored to keep the reservoir operation rules that we are simulating as simple as possible, for example based on agreed release volumes or target hydropower generation value. Operation rules based on forecasts are inevitably much more complex, which would detract from the understandability of our paper and the focus on the three Eras presented here. Regardless of whether an agreement is reached and the degree of operational complexity embedded within that agreement, the concept of the 3 eras will remain salient.
2. We are aware that it is proving to be difficult to agree upon observable hydrological quantities that will form the basis for a treaty. Even arrangements based on reservoir levels and gauged inflows to the GERD Reservoir are proving to be controversial and may be exceptionally difficult for the countries to implement. We believe that there is no realistic prospect of operational rules based on seasonal forecasts being accepted at the moment. Once an agreement based on basic hydrological variables is in place, it may be possible, in due course, to extend it to incorporate forecast information.
3. We have demonstrated in our paper that the most critical condition for the Eastern Nile Basin system will be water allocation at the end of a prolonged drought. A forecast of the drought end would be reassuring, but it will not detract from the political question that we have articulated: which reservoir should be filled first, so how should the 'pain' of the end of the drought be shared?

Comment No. 8: How realistic is a demand level for GERD to generate electricity of 1,600 MW? (is there any recent information about the current supply load and how soon new grid infrastructure might be in place?)

Authors' Response:

We used our previous studies⁴ to select this generation rate at a standard 90% reliability, and we have now expanded the paper with reference to an independent study that projects the expansion of the Electricity demand by 250% within the next decade. This projection, along with Ethiopia's own Growth and Transformation Plan and the various Power Purchase Agreements that are being negotiated, leads us to believe that demands will not fall significantly behind the supply from the GERD. Future dam developments will certainly need to re-evaluate this issue. The text now reads:

"Electrification scenarios project an increase of demands from 11.1 TWh in 2020 to 27.3 TWh in 2030²⁶, suggesting a doubling of the current 4,400 MW of installed generation capacity with the 5150 MW from the GERD could soon be justified."

Comment No. 9: It's a shame the authors don't have flow data more recent than 2002 (Figure 2), 18 years ago.... (any indication of how this missing period compares with the selected 'traces'?). More work on assessing the realism of publicly available near real-time satellite-derived rainfall estimates (e.g. CHIRPS) over the contributing basins might help with transparency for riparians and could act as

a proxy for recent flows (good rainfall - runoff relationships exist for the Blue Nile at basin-scale, and there is near real-time satellite record for the level of Lake Tana that could be helpful for extending lead times in early warning decision triggers).

Authors' Response:

We appreciate the reviewer's suggestion and have now worked to extend a naturalized sequence of flows up to 2018, which is shown in a revised Figure 2. The additional 16 years of data did not significantly change the average, minimum or maximum of the naturalized flows at Aswan, and therefore did not justify using any traces that are different from those analyzed previously.

Comment No. 10: There is a lot of work looking at implications of multi-year variability in the Nile system (not least Hurst's pioneering work on the Nile) – the Equatorial Nile and Eastern Nile systems are not strongly correlated.....'traces' or stochastic models should ideally treat them separately. What would happen if a sequence of low flow years in both systems occurred (I think White Nile flows were well above long-term average during 1978-87 and buffered the low Eastern Nile flows), or wet as was possibly the case pre-1900?

Authors' Response: We appreciate the reviewers knowledge of the Nile system and they are indeed correct that the two headwater regions are not strongly correlated. Looking at historical gauge data from Soba (Blue Nile) and Malakal (White Nile), it is apparent that the 1980's drought is primarily on the Blue Nile but the White Nile drops for a few years during this period as well. We argue that including such an analysis would confuse the narrative approach what we are advocating here, but should be considered in more detail in future research.

Comment No. 11: I think it's fair to say there is large uncertainty about short and long term flow patterns that result from multi-year drivers of variability and climate change. However, warming will continue and the already substantial evaporation losses from the reservoirs will potentially increase.....is this worth a mention (cite estimates of changes in evaporative losses from the reservoirs)?

Authors' Response: In our revised manuscript we have mentioned this issue as suggested by Reviewer #2. Specifically we added a section to the evaporation discussion in the supplemental materials as follows:

“Although this study emphasized the use of historical hydrologic scenarios to develop the narratives of the three eras, we also recognize the likely implications of climate change on the Nile Basin. Various Global Climate Models (GCMs) tend to show a wide variation in projections of precipitation, however most models tend to agree that increased temperatures are likely^{20,27}. Higher temperatures not only increase water consumption due to increased evapotranspiration²⁸, but also increase the relative advantage of storing water in less arid regions of the basin to minimize evaporative losses²⁹, thus increasing the relative benefit of storing water in the GERD Reservoir as compared to the HAD Reservoir.”

Comment No. 12: The dry period used as a trace in this paper prompted Egypt to develop capacity for

modelling and planning, a department focusing on monitoring and forecasting of Nile flows and a strong water policy focus on scarcity, but the subsequent (prolonged) return to higher flows relaxed the level of concern (Abu Zeid and Abdel-Dayem, 1992; Conway, 2005) – the GERD could act as another trigger for capacity building for DRM (and potentially cooperation) - this analysis is a timely reminder that a multi-year sequence of below average flows is a very real threat.

Authors' Response: The current situation is certainly re-emphasizing the importance of government water management capacity within the riparian countries. We have been actively involved in building that capacity. One of our key messages is that if filling the GERD proceeds uneventfully and climatic conditions permit a transition into the 'new normal' then it is possible that the "level of concern" will be "relaxed" again. Our paper argues that this would be regrettable because water management capacity will be required when a prolonged drought happens.

Reviewer #3

25 – It is important to extend the average flow of the Nile to cover the period 2002 up to 2019.

Authors' Response:

We appreciate the reviewer's suggestion and have worked to extend of the sequence of naturalized flows until 2018, which is shown in a revised Figure 2 (also mentioned in our response to Reviewer #2.) The additional 16 years of data did not significantly change the average, minimum or maximum of the naturalized flows at Aswan, and therefore did not justify using traces that are different from those analyzed previously.

34. HAD total design capacity was 162 bcm ad dead storage was 31.6 bcm. However, due to reservoir siltation of 7 bcm, the capacities are now different, 46% is deposited in live storage.

47. HAD design capacity was 162 bcm, now 155 bcm.

Authors' Response: We appreciate the reviewer's suggestion and have made this adjustment to our results, to account for sedimentation. We also added the following statement:

"Since construction of the dam, sediment accumulation has reduced the total capacity by approximately 7 bcm, with approximately 46% deposited in the live storage."

60. GERD new capacity of turbines is 6450 MW and hence annual power generated will be in the range of 15,962-17,500 GWh.

Authors' Response: We thank you for this detail. We acknowledge that the installed capacity stated by Sudan once reached 6450 MW, however we now understand this was subsequently reduced by the Ethiopian Ministry of Water, Irrigation and Energy in 2019. Based on the statement on 14 May 2020 by the Govt of Ethiopia to the UN Security Council³⁰, we have updated the installed capacity to 5150 MW.

141. There is no logic to build the assumptions on HAD storage of 120 bcm. It is well known that the first or initial filling is going to start on July 2020 where the HAD water level will be 179.6 m, or a bit less, with a capacity of 147 bcm. The paper has to work on actual measured values.

Authors' Response:

We appreciate the reviewer's comments and his encouragement to update the model runs. As recommended by Reviewer #3, we have adjusted the modeling runs for Era 1 (Filling) to reflect the most recent pool elevations derived satellite measurements. The logic for Eras 2 and 3 is to start the HAD Reservoir partially full because no one knows where the reservoir storage will be or what the condition will be when a 'new normal' or 'extended drought' era begins. It would be unrealistically optimistic to start Eras 2 and 3 (which correspond to some unknown future starting date) with the reservoir full.

149. Figure 4. Use the actual situation (level and volume of water) expected to be at HAD on the first of July 2020.

Authors' Response:

As described above, we have updated the initial condition to represent the actual pool elevation of the HAD at the 2020 start time of the model.

456 – Sudan total abstraction is 12.2 bcm at Aswan and not 16.7. This is based on actual measured values.

469-471 – Diversion of Sudan is measured by both Sudan and Egypt authorities and not debated. This is based on the Permanent Joint Technical Commission, PJTC, record.

472 – Assuming 16.7 bcm withdrawal by Sudan is not right.

Authors' Response:

To respond to Reviewer #3's concern about our estimate of Sudanese abstraction measured at Aswan of 16.7 bcm, in the Supplementary Materials we have included a sensitivity analysis that shows the consequences of our results for an assumed Sudanese withdrawal of 12.2 bcm measured at Aswan. Nonetheless, the main analysis in the paper uses the higher estimate. Here we would note that many knowledgeable observers believe that total abstractions by Sudan are substantially above 12.2 bcm measured at Aswan. Nearly 15 years ago Sudan provided Blackmore and Whittington an estimate total abstraction by Sudan at 13.8 bcm, and this did not include evaporation losses from the Merowe Reservoir. Most of the modelers working in the Nile basin are now using estimates of total abstraction by Sudan close to our estimate of 16.7 bcm, relying on data collected from various parties in the basin^{4,10,31,32}.

474 – Assuming withdrawal of 2 bcm from lower Beles is exaggerated since GERD reservoir is going to submerge 80-90 km of the arable land in this part of the lower Beles.

Authors' response: We appreciate the clarification from the reviewer about this physical reality. The additional withdrawals have been modified to only show the proportion reported for the Upper Beles Command Area, transfers from the Beles to the Dinder and irrigation projects in the Arjo and Anger sub-basins. We added a specific citation providing the source of these estimates.

547 – Roseries reservoir capacity is now 5.9 bcm.

Authors' response: We have made this clarification now in the revised manuscript.

709 – Table S1. Roseries minimum pool elevation is now 471 instead of former level of 467m. Also review HAD levels.

Authors' response: We appreciate this updated information and have incorporated this into the updated manuscript and analysis.

712 – Table S2. Sudan water demand is now 12.2 bcm measured at Aswan. Ethiopia water demand to be also reviewed on what is mentioned in 474.

Authors' response: see our answers above.

730. Table S4, and Table S5. Review Sudan annual evaporation based on Merowi 1550 m, Sennar 300 m, and Roseries 650 m. Review also HAD evaporation.

Authors' response: After making the requested updates to the initial conditions of the model, we have now revised all of the results in the supplementary tables including the evaporation volumes, which are specific to the particular scenarios evaluated in the model.

Conclusion ...

- 1. This paper overlooked that the existing situation in HAD reservoir is a combination of the unutilized share of Sudan quota (6 bcm at Aswan) and the extra flood waters accumulated during the last three decades that should also be divided among the two countries (3.5 bcm for each) according to the 1959 Nile water Agreement.**

Authors' response: We acknowledge the HAD has refilled to be almost full today, and Figure S3 shows the most recent storage levels as of 15 June 2020. Indeed, this is likely the result of relatively elevated Nile flows and upstream development that has been lower than anticipated, allowing Egypt to make releases in excess of the allocation specified in the 1959 agreement. In this paper we focus on the three eras as future and expected conditions that the countries will face, and we explicitly do not evaluate the legal interpretations of how previously accumulated excess water should now be allocated. We believe this is outside of the scope of this paper.

- 2. Evaluation of the first filling has to start from the actual level and capacity of HAD on first of July 2020 not based on a hypothetical level and capacity.**

Authors' response: We appreciate this suggestion and have updated our analysis to reflect the current storage levels in the HAD.

- 3. The paper is timely and covering a pertinent issue now under consideration and negotiation by the three countries.**

Authors' response: We appreciate Reviewer #3's assessment that our paper is timely and important.

- 1 Arjoon, D., Mohamed, Y., Goor, Q. & Tilmant, A. Hydro-economic risk assessment in the eastern Nile River basin. *Water Resources and Economics* **8**, 16-31, doi:10.1016/j.wre.2014.10.004 (2014).
- 2 Goor, Q., Halleux, C., Mohamed, Y. & Tilmant, A. Optimal operation of a multipurpose multireservoir system in the Eastern Nile River Basin. *Hydrol. Earth Syst. Sci.* **14**, 1895-1908, doi:10.5194/hess-14-1895-2010 (2010).
- 3 Arjoon, D., Tilmant, A. & Herrmann, M. Sharing water and benefits in transboundary river basins. *Hydrol. Earth Syst. Sci.* **20**, 2135-2150, doi:10.5194/hess-20-2135-2016 (2016).
- 4 Wheeler, K. G. *et al.* Exploring Cooperative Transboundary River Management Strategies for the Eastern Nile Basin. *Water Resources Research* **54**, 9224-9254, doi:doi:10.1029/2017WR022149 (2018).
- 5 Block, P. J. & Strzepek, K. Economic analysis of large-scale upstream river basin development on the Blue Nile in Ethiopia considering transient conditions, climate variability, and climate change. *Journal of Water Resources Planning and Management* **136**, 156-166, doi:10.1061/(ASCE)WR.1943-5452.0000022 (2010).
- 6 King, A. & Block, P. An assessment of reservoir filling policies for the Grand Ethiopian Renaissance Dam. *Journal of Water and Climate Change* **5**, 233-243, doi:10.2166/wcc.2014.043 (2014).
- 7 Zhang, Y., Block, P., Hammond, M. & King, A. Ethiopia's Grand Renaissance Dam: Implications for Downstream Riparian Countries. *Journal of Water Resources Planning and Management*, doi:10.1061/(ASCE)WR.1943-5452.0000520 (2015).
- 8 Zhang, Y., Erkyihum, S. T. & Block, P. Filling the GERD: evaluating hydroclimatic variability and impoundment strategies for Blue Nile riparian countries. *Water International* **41**, 593-610, doi:10.1080/02508060.2016.1178467 (2016).
- 9 Jeuland, M. Economic implications of climate change for infrastructure planning in transboundary water systems: An example from the Blue Nile. *Water Resources Research* **46**, W11556, doi:10.1029/2010WR009428 (2010).
- 10 Jeuland, M. & Whittington, D. Water resources planning under climate change: Assessing the robustness of real options for the Blue Nile. *Water Resources Research* **50**, 2086-2107, doi:10.1002/2013WR013705 (2014).
- 11 Jeuland, M., Wu, X. & Whittington, D. Infrastructure development and the economics of cooperation in the Eastern Nile. *Water International*, 1-21, doi:10.1080/02508060.2017.1278577 (2017).
- 12 Nigatu, G. & Dinar, A. Economic and hydrological impacts of the Grand Ethiopian Renaissance Dam on the Eastern Nile River Basin. *Environment and Development Economics* **21**, 532-555, doi:10.1017/S1355770X15000352 (2015).
- 13 Kahsay, T. N. *et al.* A hybrid partial and general equilibrium modeling approach to assess the hydro-economic impacts of large dams – The case of the Grand Ethiopian Renaissance Dam in the Eastern Nile River basin. *Environmental Modelling & Software* **117**, 76-88, doi:<https://doi.org/10.1016/j.envsoft.2019.03.007> (2019).
- 14 Borgomeo, E., Mortazavi-Naeini, M., Hall, J. W. & Guillod, B. P. Risk, Robustness and Water Resources Planning Under Uncertainty. *Earth's Future* **6**, 468-487, doi:10.1002/2017ef000730 (2018).

- 15 Borgomeo, E., Mortazavi-Naeini, M., Hall, J. W., O'Sullivan, M. J. & Watson, T. Trading-off tolerable risk with climate change adaptation costs in water supply systems. *Water Resources Research* **52**, 622-643, doi:10.1002/2015WR018164 (2016).
- 16 Harding, B. L., Sangoyomi, T. B. & Payton, E. A. Impacts of a severe sustained drought on Colorado River water resources. *JAWRA Journal of the American Water Resources Association* **31**, 815-824, doi:10.1111/j.1752-1688.1995.tb03403.x (1995).
- 17 Georgakakos, A. P. *et al.* Value of adaptive water resources management in Northern California under climatic variability and change: Reservoir management. *Journal of Hydrology* **412-413**, 34-46, doi:<https://doi.org/10.1016/j.jhydrol.2011.04.038> (2012).
- 18 Maass, A. *Design of water-resource systems : new techniques for relating economic objectives, engineering analysis, and governmental planning.* (Harvard University Press., 1962).
- 19 Hurst, H. E., Black, R. P. & Simaika, Y. M. *Long-term storage: an experimental study.* (Constable, 1965).
- 20 Conway, D. From headwater tributaries to international river: Observing and adapting to climate variability and change in the Nile basin. *Global Environmental Change* **15**, 99-114, doi:<http://dx.doi.org/10.1016/j.gloenvcha.2005.01.003> (2005).
- 21 Ward, N. & Conway, D. Applications of interannual-to-decadal climate prediction: An exploratory discussion on rainfall in the Sahel region of Africa. *Climate Services* **18**, 100170, doi:<https://doi.org/10.1016/j.cliser.2020.100170> (2020).
- 22 Sutcliffe, J., Hurst, S., Awadallah, A. G., Brown, E. & Hamed, K. Harold Edwin Hurst: the Nile and Egypt, past and future. *Hydrological Sciences Journal* **61**, 1557-1570, doi:10.1080/02626667.2015.1019508 (2016).
- 23 Koutsoyiannis, D. Hydrology and change. *Hydrological Sciences Journal* **58**, 1177-1197, doi:10.1080/02626667.2013.804626 (2013).
- 24 Koutsoyiannis, D., Yao, H. & Georgakakos, A. Medium-range flow prediction for the Nile: a comparison of stochastic and deterministic methods / Prévision du débit du Nil à moyen terme: une comparaison de méthodes stochastiques et déterministes. *Hydrological Sciences Journal* **53**, 142-164, doi:10.1623/hysj.53.1.142 (2008).
- 25 in *Congressional Bill 2030 (S. 1057)* (116th Congress, Public Law 116-14, USA, 2019).
- 26 Mondal, M. A. H., Bryan, E., Ringler, C., Mekonnen, D. & Rosegrant, M. Ethiopian energy status and demand scenarios: Prospects to improve energy efficiency and mitigate GHG emissions. *Energy* **149**, 161-172, doi:<https://doi.org/10.1016/j.energy.2018.02.067> (2018).
- 27 Conway, D. Water resources: Future Nile river flows. *Nature Clim. Change* **7**, 319-320, doi:10.1038/nclimate3285 (2017).
- 28 Hasan, E., Tarhule, A., Kirstetter, P.-E., Clark, R. & Hong, Y. Runoff sensitivity to climate change in the Nile River Basin. *Journal of Hydrology* **561**, 312-321, doi:<https://doi.org/10.1016/j.jhydrol.2018.04.004> (2018).
- 29 Wang, W. *et al.* Global lake evaporation accelerated by changes in surface energy allocation in a warmer climate. *Nature Geoscience* **11**, 410-414, doi:10.1038/s41561-018-0114-8 (2018).
- 30 Government of Ethiopia. Vol. S/2020/409 (United Nations Security Council, United Nations Digital Library, 2020).

- 31 van der Krogt, W. & Ogink, H. Development of the Eastern Nile Water Simulation Model. Report No. 1206020-000-VEB-0010, (Deltares, Delft, 2013).
- 32 Multsch, S. *et al.* Improving irrigation efficiency will be insufficient to meet future water demand in the Nile Basin. *Journal of Hydrology: Regional Studies* **12**, 315-330, doi:<https://doi.org/10.1016/j.ejrh.2017.04.007> (2017).

REVIEWERS' COMMENTS:

Reviewer #2 (Remarks to the Author):

The authors have done a good job in responding to the referee comments and revising the paper. The justification of the research significance is convincing (although it could be signalled more in the introduction).

This is a very clearly written and accessible analysis coming at a crucial time for the Nile and its publication will hopefully provide an excellent benchmark for further studies and debates on this contentious issue.

I recommend publication without any further revision.

I have one observation, that does not need addressing, re the description of 'consequences of a severe multi-year drought' (from Line 337). Clearly there is major concern about how low HAD reservoir levels will be felt/perceived in Egypt, however, it's worth noting that for Ethiopia, with almost total reliance on rainfed agriculture in the Blue Nile source areas, food security is very likely to be a major concern (and possibly politically destabilising) during a multi-year low rainfall event. This could further hamper reasoned political decision-making.

Reviewer #3 (Remarks to the Author):

I reviewed the responses of the authors. The only difference is on the issue of Sudan abstraction. It is important to differentiate between Sudan abstraction and water balance which includes reservoirs evaporations. This is very sensitive for Sudan. They have to separate Sudan abstraction from reservoir evaporation.